# A Novel Orthotopic Liver Cancer Model for Creating a Human-like Tumor Microenvironment

**DOI:** 10.3390/cancers13163997

**Published:** 2021-08-08

**Authors:** Rong Qiu, Soichiro Murata, Chao Cheng, Akihiro Mori, Yunzhong Nie, Satoshi Mikami, Shunsuke Hasegawa, Tomomi Tadokoro, Satoshi Okamoto, Hideki Taniguchi

**Affiliations:** 1Department of Regenerative Medicine, Yokohama City University Graduate School of Medicine, 3-9, Fuku-ura, Kanazawa-ku, Yokohama, Kanagawa 236-0004, Japan; t186027c@yokohama-cu.ac.jp (R.Q.); t176065b@yokohama-cu.ac.jp (A.M.); smik@yokohama-cu.ac.jp (S.M.); s3hasegawa@yokohama-cu.ac.jp (S.H.); tadokoro@yokohama-cu.ac.jp (T.T.); sokamoto@yokohama-cu.ac.jp (S.O.); 2Division of Regenerative Medicine, University of Tokyo, 4-6-1, Shirokanedai, Minato-ku, Tokyo 108-8639, Japan; yznie@ims.u-tokyo.ac.jp; 3Department of Nuclear Medicine, Shanghai Changhai Hospital, 168 Changhai Rd, Shanghai 200433, China; 13501925757@163.com

**Keywords:** hepatocellular carcinoma, animal model, tumor microenvironment, fibrosis, ultra-purified alginate gel

## Abstract

**Simple Summary:**

Hepatocellular carcinoma is the most common form of liver cancer. The lack of models that resemble actual tumor development in patients, limits the research to improve the diagnosis rate and develop new treatments. This study describes a novel mouse model that involves organoid formation and an implantation technique. This mouse model shares human genetic profiles and factors around the tumor, resembling the actual tumor development in patients. We demonstrate the roles of different cell types around the tumor, in promoting tumor growth, using this model. This model will be useful to understand the tumor developmental process, drug testing, diagnosis, prognosis, and treatment development.

**Abstract:**

Hepatocellular carcinoma (HCC) is the most common form of liver cancer. This study aims to develop a new method to generate an HCC mouse model with a human tumor, and imitates the tumor microenvironment (TME) of clinical patients. Here, we have generated functional, three-dimensional sheet-like human HCC organoids in vitro, using luciferase-expressing Huh7 cells, human iPSC-derived endothelial cells (iPSC-EC), and human iPSC-derived mesenchymal cells (iPSC-MC). The HCC organoid, capped by ultra-purified alginate gel, was implanted into the disrupted liver using an ultrasonic homogenizer in the immune-deficient mouse, which improved the survival and engraftment rate. We successfully introduced different types of controllable TME into the model and studied the roles of TME in HCC tumor growth. The results showed the role of the iPSC-EC and iPSC-MC combination, especially the iPSC-MC, in promoting HCC growth. We also demonstrated that liver fibrosis could promote HCC tumor growth. However, it is not affected by non-alcoholic fatty liver disease. Furthermore, the implantation of HCC organoids to humanized mice demonstrated that the immune response is important in slowing down tumor growth at an early stage. In conclusion, we have created an HCC model that is useful for studying HCC development and developing new treatment options in the future.

## 1. Introduction

Liver cancer is one of the leading causes of cancer deaths globally, and accounts for approximately 810,000 deaths annually, making it a major challenge for the global healthcare system [1]. Between 1990 and 2015, the incidence rate of liver cancer has increased by 75% worldwide. It is predicted that the number might further increase, due to worldwide socio-economic changes [1]. Hepatocellular carcinoma (HCC) is the most common form of liver cancer [2,3]. HCC is mainly caused by chronic liver diseases, such as chronic liver inflammation disease, liver fibrosis/cirrhosis, caused by hepatitis B virus and hepatitis C virus infection, alcohol consumption, non-alcoholic fatty liver disease, and non-alcoholic steatohepatitis [4]. Although there are usually no obvious physical symptoms at the early stages of HCC, only a small portion of HCC patients can be diagnosed early. The majority of the patients are diagnosed at the advanced stage, leaving limited treatment options [5]. Considering that the prognosis of HCC remains very poor, several researchers are focusing on improving early diagnosis methods and developing new therapeutic options. To achieve these, first, we may need to have a better understanding of HCC pathogenesis.

Animal models have been long used to study cancer pathogenesis, and as a tool for drug screening. Because of the complicated etiology of cancer, the heterogeneity of cancer cells, and the tumor microenvironment (TME), it is very difficult to develop an animal model that can sufficiently mimic the human cancer disease process. Currently, there are a few types of HCC animal models, i.e., (1) genetically modified models, (2) chemically induced models, and (3) xenograft models. Each animal model type has its advantages and limitations. Genetically modified HCC models [6,7,8], such as the c-Myc/Tgfα transgenic mouse model, have the significant advantages of rapid reproductive capacity, longer life span, and consistent cancer physiology. However, the major limitations of these genetically modified models are the different genetic profiles compared to humans and the relatively low tumor mutation rates. Moreover, the spontaneous HCC in these animals is not induced by common HCC stimulants, such as the fibrosis or cirrhosis microenvironment, making whether these models can mimic HCC pathogenesis in humans questionable. In the chemically induced models, chemicals are introduced on the animal to cause liver damage and eventually induce HCC. Common chemically induced animal models include inducing murine with diethylnitrosamine and carbon tetrachloride [9]. The chemically induced models are good for the pathological observation of HCC development; however, it generally takes a long period for tumor induction, and the genetic background of the tumor is undefined. Xenograft models are usually generated by introducing human cancer cell suspension to immune-deficient animals, through subcutaneous injection. Although there are some xenograft models that are generated through orthotopic implantation by intra-hepatic injection or intra-splenic injection, these methods often cause vascular embolism in the animals. Xenograft models have the advantages of having the same genetic profile as humans. Nevertheless, due to the lack of the original hepatic TME, it is difficult to accurately reflect the real clinical process. In addition, the preclinical drug screening trials in animal models have shown that many drugs are not sensitive to the patients [10]. There is an urgent need to establish novel, reliable, and robust HCC animal models that can be easily and rapidly produced, carry HCC with the same genetic profiles as human HCC, and mimic the normal and pathological liver TME in humans. This model will be a useful tool for HCC pathogenesis and therapeutic studies.

Tumors develop in a complicated tissue environment, called the TME, which consists of tumor cells in an extracellular matrix, as well as a collection of stromal cells, such as fibroblast [11,12], tumor-associated macrophages [13], endothelial cells (EC) [14], and mesenchymal cells (MC) [15]. These cells interact closely with each other and the surrounding tissue, to create a microenvironment that has the optimal physical, molecular, and cellular factors that support tumor growth [16]. Recently, three-dimensional (3D) tumor models, consisting of tissue-specific tumor spheroids and organoids, have been used for studying cancer development, metastasis, and the TME [17]. Tumor spheroids and organoids are generated by the co-culture of stromal cells with biomimetic scaffold materials, such as alginate, collagen, gelatin, hyaluronic acid, and Matrigel [17,18,19,20,21]. The 3D spheroid and organoid models are changing the paradigm of cancer research, because they can resemble the dynamic and complicated TME during clinical cancer development, unlike traditional bi-dimensional (2D) cell culture models [22]. The TME plays a significant role in tumor recurrence, promoting tumor recurrence in about 70% of patients who received resection or local ablation [23]. Several important findings about the TME mechanisms, tumor development, metastasis, and drug screening, have been reported using tumor spheroids/organoids models [17,18,19,20,21,24]. Liu et al. demonstrated that mesenchymal stem cells enhance the metastasis of HCC cells via TGF-β in vitro, a phenomenon which is difficult to test using 2D cell culture models [18]. Han et al. observed that tumorigenicity markers in the human lung cancer spheroids were enhanced when co-cultured with human mesenchymal stem cells over the chitosan–hyaluronan coating plate in vitro. The tumor was also confirmed in the head, anterior trunk, and posterior trunk, by injected a 3D co-culture cancer spheroid platform in the zebrafish embryos [20]. Sasser et al. proved that stromal cells, co-cultured with Matrigel, upregulated the breast cancer cell growth rates in the TME [21]. Also, the 3D co-cultured cancer spheroid platform or organoid has been ectopically implanted beneath the skin [25], or the kidney sub-capsules or spleens, in the immunodeficient mouse, to induce the tumor [26]. Since the ectopic environment is different from its native environment, it may ultimately affect the tumorigenesis [27]. Orthotopic implantations, such as the needle cecal injection of colorectal cancer cell lines, induced tumorigenesis and distant metastasis [28]. However, the needle orthotopic injection can cause blood vessel embolism and increase mortality. Furthermore, some researchers have reported that the needle orthotopic injection of human and mouse colorectal cancer organoids, did not create cancer [29]. Recently, 3D spheroid and organoid models of the lung, prostate, and breast, generated using stromal cells, have been reported [20,21,27], and the range of available models is expected to keep growing.

We have previously reported the first in vitro grown functional human liver bud (LB) organoids, generated from differentiated cells derived from human iPSC, known as iPSC-LB [30,31,32]. The iPSC-LB revealed resemblances between iPSC-LB and ED 10.5 murine fetal livers. This observation inspired us to modify the iPSC-LB technology, and create novel HCC animal models that are robust and rapidly produced, carry tumors with similar genetic profiles as humans, and possess adjustable TMEs of HCC.

Compared to the iPSC-LB, a single miniature organoid [30,31,32], the new method merges several miniature organoids into a sheet-like organoid structure. The sheet-like organoid can be implanted in vivo. A functional sheet-like human HCC organoid was generated in vitro and implanted into the livers of healthy mice, using ultra-purified alginate (UPAL) gel [33]. This study successfully improved our previous iPSC-LB organoid method, and we applied it to the generation of a novel human HCC mouse model, within two weeks. Immunostaining analyses revealed that the HCC in these mice had similar pathological profiles as human HCC tumors. Additionally, our HCC mouse model is useful in testing the effects of the TME on the development of HCC.

## 2. Materials and Methods

### 2.1. Cell Lines

Human HCC (Huh7, HepG2), human iPSC (Ff-I01s04), and human pancreatic cancer cells (CFPAC-1) were used in this study. Huh7 (RCB1366) and HepG2 (RCB1648) were purchased from RIKEN BRC Cell Bank (Tsukuba, Ibaraki, Japan). The 293FT cell line was purchased from Thermo Fisher Scientific Corp (R70007). Human iPSC line, Ff-I01s04 was provided by CiRA, Kyoto University (Dr. Shinya Yamanaka). CFPAC-1 (CRL-1918™) was obtained from American Tissue Culture Collection.

### 2.2. Establishment of Stable Luciferase-Expressing Huh7 Cells (Huh7-Luci)

Huh7 cells were cultured in Dulbecco’s modified Eagle’s medium (DMEM; Wako, Osaka, Japan) containing 10% fetal bovine serum (FBS) at 37 °C in 5% CO_2_. Plasmids, pLenti-luciferase-P2A-Neo (Cat# 05621), and pCMV-VSV-G (Cat# 8454) were purchased from Addgene (https://www.addgene.org/ (accessed on 30 October 2019)) (Watertown, MA, USA). Plasmid pLetni-luciferase-P2A-Neo and pCMV-VSV-G were co-transfected to 293FT with polyethyleneimine (PEI) as previously described [34,35] to construct pseudotype luciferase-expressing lentivirus. Further, 293FT is a derivative of the 293T cells transformed with the SV40 large T antigen. Forty-eight hours post-transfection, the pseudotype virus was harvested from the cell supernatant followed by filtration with a 0.2 μm disposable membrane filter (Toyo Roshi Kaisha, Ltd., Tokyo, Japan). Huh7 cells (10^6^ cells/well) with 0.3 MOI (multiplicity of infection) of the luciferase-expressing VSV-G pseudotype virus and 1 mg/mL of Polybrene (Santa Cruz Biotechnology, Dallas, TX, USA) in a 12-well plate were centrifuged at 1000× *g* for 2 h at 33 ℃, then cultured in a 37 ℃ incubator with 5% CO_2_ for 24 h. The cells were selected with G418 for 2 weeks.

### 2.3. Generation of the HCC Organoid

Huh7-Luci and Ff-I01s04 human iPSC were used to generate HCC organoids. Ff-I01s04 cells were maintained on Laminin 511 E8 fragment (iMatrix-511™, provided by Nippi)-coated dishes in StemFit AK02N culture medium (Ajinomoto, Tokyo, Japan). The Ff-I01s04 cells were used to generate iPSC-EC and iPSC-MC as described previously [30]. Huh7-Luci, iPSC-EC and iPSC-MC were dissociated using Gibco™ trypsin-EDTA (0.25%), and 100 μL of cell suspension at a ratio of 10:2:2 (Huh7-luci:iPSC-EC:iPSC-MC) was placed into each well of an Ibidi culture-insert 2 well system (Ibidi GmbH, Gräfelfing, Germany) in a six-well plate. Huh7-Luci cells were seeded into each well of the Ibidi culture-insert 2 well system at a density of 5 × 10^5^ cells/unit. The HCC organoid culture medium is a mixture of DMEM medium supplemented with 0.1 μM dexamethasone (Sigma-Aldrich, St. Louis, MO, USA) and 10 ng/mL oncostatin M (R&D Systems, Minneapolis, MN, USA) with an equal volume of KBM VEC-1 medium (Kohjin Bio, Saitama, Japan) containing KBM VEC-1 supplement. The HCC organoid was formed spontaneously after one culture day. After six days, the HCC organoid was implanted in vivo. ECs were infected with retroviruses expressing the genes encoding Kusabira-Orange (KO), as described previously [32,36] for live-cell imaging. The living cell images were observed under a BZ-X710 all-in-one fluorescence microscope (Keyence, Osaka, Japan).

### 2.4. Generation of the Pancreatic Cancer Organoid

CFPAC-1, iPSC-EC, and iPSC-MC were used to generate the pancreatic cancer organoids. The CFPAC-1 cells were maintained in Iscove’s modified Dulbecco’s medium (Gibco, Grand Island, NE, USA) supplemented with 10% FBS. Ff-I01s04 cells were used to generate iPSC-EC and iPSC-MC as described previously [30]. CFPAC-1, iPSC-EC, and iPSC-MC were dissociated from their culture plates with Gibco™ trypsin-EDTA (0.25%), a CFPAC-1/iPSC-EC/iPSC-MC cell suspension mixture (10:2:2) was prepared and added to the Ibidi culture-insert 2 well system (100 μL/well) in the six-well plate. CFPAC-1 cells (4 × 10^5^) were seeded in each well of the Ibidi culture-insert 2 well system. The pancreas cancer organoid medium was a mixture of DMEM medium containing dexamethasone and oncostatin M, mixed with an equal amount of KBM VEC-1 medium containing KBM VEC-1 supplement. After two days, the pancreatic cancer organoid was used to implant in vivo experiment.

### 2.5. Animal

Male, 7-week-old non-obese diabetic/Shi-scid, IL-2RγKO Jic (NOG) mice (In-Vivo Science Inc., Tokyo, Japan), female humanized mice [37], NOG-HLA-A2 transgenic mice (In-Vivo Science Inc., Tokyo, Japan), female 10-week-old NOD-SCID mice (Sankyo Labo Service Corporation, Inc. (Tokyo, Japan)) were used in this study. All animal experiments were conducted following the ethics regulations established by the Animal Experiment Committee of Yokohama City University, and the Animal Experiment Committee approved the animal experiment methods (approval No. 17-025). All mice were acclimatized in the animal center of Yokohama City University for 1 week before the experiment.

### 2.6. HCC Organoid Implantation

Eight-week-old NOG mice and 11-week-old NOD-SCID mice were used. All mice were anesthetized with isoflurane before implantation. The animal’s abdomen was first excised longitudinally to expose the internal organs of the abdominal cavity. Then, an ultrasonic homogenizer (UH; Yamato Scientific, Tokyo, Japan) was used to disrupt the surface of the median lobe in the mouse liver. The depth of the median lobe that was disrupted by UH was 2 mm. The surface of the median lobe was pressed with anti-bleeding gauze to stop bleeding. Next, the HCC organoids were implanted onto the surface of the median lobe in the mouse liver. After that, 0.5% UPAL (Mochida Pharmaceutical Co. Ltd. Tokyo, Japan) [33] was applied over the implanted HCC organoids. At last, 10% calcium chloride was added over the UPAL to promote the gelation of UPAL. The amount of capping agent applied was equal in volume with the implantation site (through estimation).

### 2.7. Pancreatic Cancer Organoid Implantation

Eight-week-old NOG mice were used for this experiment. All mice were anesthetized by isoflurane. The animal’s left abdomen was first excised transversely to expose the pancreas. Then, a UH was used to sharpen the surface of the pancreas body. The surface of the pancreas body was pressed with anti-bleeding gauze to stop bleeding. Next, the pancreas cancer organoids were implanted onto the surface of the pancreas body. After that, 0.5% UPAL was applied over the implanted pancreas cancer organoids. Finally, 10% calcium chloride was added over the UPAL.

### 2.8. Induction of Liver Fibrosis in Mice

Liver fibrosis was induced in the 11-week-old NOD-SCID mice by injection of thioacetamide (TAA) (Sigma-Aldrich, Darmstadt, Germany) into the peritoneal cavity (100 mg/kg) 3 times per week, for four consecutive weeks.

### 2.9. Induction of Non-Alcoholic Fatty Liver Disease (NAFLD) in Mice

Eleven-week-old NOD-SCID mice were fed with a choline-deficient L-amino acid-defined high-fat diet (CDAHFD) (Research Diets Inc., New Brunswick, NJ, USA) for five consecutive weeks to induce the NAFLD condition.

### 2.10. Oil Red O (ORO) Staining

The frozen slides of tissue were air-dried for one hour at room temperature. Slides were washed with Milli-Q ultrapure water for 30 s and incubated with 60% 2-propanol for 1 min. Next, slides were stained with Oil Red O solution (Sigma-Aldrich, Darmstadt, Germany) for 15 min. Then, immersed with 60% 2-propanol for 1 min again, washed with Milli-Q ultrapure water for 30 s. After that, incubated with hematoxylin for 15 min, and washed with running water 10 min before imaging with an Olympus VS120 high-throughput automated slide scanning microscope.

### 2.11. Hematoxylin and Eosin (HE) Staining

Firstly, the slides were deparaffinized by xylene for 15 min and then gradually dehydrated from 100% to 50% ethanol solution. Next, the tissues were stained with hematoxylin solution for 20 min at room temperature and rinsed with running water for nearly 10 min. Secondly, eosin was used to immerse the tissue for 15 min, and the slides were rinsed with Milli-Q ultrapure water for 5 min. Finally, these slides were performed with 50%, 70%, 100% ethanol solution two times for 2 min and then in xylene three times for 3 min.

### 2.12. Immunofluorescence Staining

Paraffin sections were cut at 7 μm in each slide. The deparaffinization method is similar to the previous HE staining protocol. Staining dish containing antigen retrieval solution (10 mM sodium citrate buffer) was placed in the autoclave until the temperature reached 121 °C for 20 min. The staining dish was removed at room temperature and these slides were cooled for 50 min. Paraffin-embedded tissue sections form protein cross-links that hide the antigenic sites in tissue specimens, and therefore provide weak or false negative immunofluorescence staining results. The addition of 10 mM sodium citrate buffer in the autoclave breaks the protein cross-links, thereby showing the antigens and epitopes in the paraffin-embedded tissue sections, to intensify the antibody staining. Sections were rinsed in 0.05% PBS–Tween 20 three times for 5 min. The slides were soaked in protein blocking solution (Cat# 1114053, DAKO) for 1 h at room temperature. Slides were incubated with primary antibody at appropriate dilution in protein block solution overnight at 4 °C. The following antibodies were used for this study: anti-human AFP (Cat# GA500, DAKO), anti-human CD31 (Cat# M0823, DAKO), anti-human/mouse Vimentin (Cat# MAB21052, R&D Systems), and anti-human ki67 (Cat# M7240, DAKO). Slides were rinsed with 0.05% PBS–Tween 20 for 3 × 5 min. Then, a secondary antibody was added near 1 h at room temperature. Slides were rinsed with 0.05% PBS–Tween 20 for 3 × 5 min. Finally, DAPI at appropriate dilution was used to stain the nucleus for 10 min.

### 2.13. Sirius Red Staining

The steps of deparaffinization are similar to the previous HE staining method. The slides were applied in 0.03% Sirius red–picric acid solution to cover the slides and incubate for 60 min completely. Slide were rinsed quickly with 0.5% acetic acid solution two times for 2 min. After that, the slides were washed by Milli-Q ultrapure water for 5 min. Finally, dehydrated in 50%, 70%, 100% ethanol solution two times for 2 min and then in xylene three times for 3 min.

### 2.14. Extraction of the Total Lipid Fraction from Liver

Approximately 20 mg of tissue was sampled from the thawed middle lobe of the liver. Excess water was absorbed with filter paper. Next, samples were weighed precisely. Each sample was mechanically homogenized in an extraction buffer consisting of chloroform/methanol (1:2) solution (volume of the extraction solution is ten times the tissue weight) with a masher (power masher, Nippi, Tokyo, Japan). The chloroform/methanol solution (the chloroform/methanol solution is 30 times the tissue weight) was added. After the vortex, 6N hydrochloric acid was added (volume of the 6N hydrochloric acid solution is 0.2 times the tissue weight). Next, the chloroform solution was added (chloroform solution is eight times the tissue weight), then centrifuged at 2000 rpm for 5 min to collect the lower layer of liquid. Finally, the chloroform/methanol/0.1 N hydrochloric acid (3:48:47) (chloroform/methanol/0.1 N hydrochloric acid solution is 10 times the tissue weight) was added then centrifuged at 2000 rpm for 5 min. The lower layer of liquid was collected to determine the liver fat weight. The triglyceride (TG) content in total lipid extract was determined using the LabAssay^TM^ triglyceride assay kit (Wako Chemicals, Osaka, Japan), following the manufacturer’s instructions.

### 2.15. Human Albumin (ALB) Concentration in Culture Medium

Human ALB was measured by using the human ALB ELISA kit (Bethyl Laboratories, Montgomery, TX, USA) according to the manufacturer’s instructions.

### 2.16. Luciferase Assay In Vitro

HCC organoid was homogenized by a masher in a 1.5 mL tube with 200 μL of cell culture lysis reagent. After being centrifuged at 2000× *g* for 3 min, the supernatant was used for the luciferase assay. 4 μL of cell lysate was mixed with 20 μL of luciferase assay substrate (luciferase assay system, Promega, Madison, WI, USA). The intensities of luciferase fluorescence emission of Huh7-Luci were measured by a luminometer (PerkinElmer).

### 2.17. In Vivo Bioluminescence Imaging (BLI)

D-luciferin (Promega, Madison, WI, USA) was injected intraperitoneally into mice at 150 mg/kg before BLI. Mice were placed in the imaging chamber under isoflurane anesthesia. All images were taken 5 min after D-luciferin injection. Photon emission from the regions of interest was measured using an IVIS 200 device (Xenogen Corp., Alameda, CA, USA).

### 2.18. Statistical Analysis

Statistical analyses were performed by using GraphPad Prism 8 software. For studies comparing two groups, we applied the Mann–Whitney *U* test. For studies with multiple groups comparison, one-way ANOVA was applied. All data were presented as the mean ± standard error of the mean. Differences were considered significant if the *p*-value was <0.05 (*), <0.01 (**), <0.001 (***), or <0.0001 (****).

## 3. Results

### 3.1. Generation of HCC Organoid from Huh7-Luci Co-Cultured with EC plus MC Derived from Human iPSC

Previously, our group had reported a method to generate mini LB organoids in vitro [30,31]. In the current study, we modified and improved this method to generate a functional HCC organoid for implantation. Huh7-Luci, iPSC-EC, and iPSC-MC cells were co-cultured and induced to form HCC organoids (Materials and Methods Section 2.3.). A combination of these HCC organoids in the Ibidi culture-insert 2 well system after six days, will lead to the formation of a complete 3D HCC organoid that can be used for implantation. The area of each mini HCC organoid is 0.08 mm^2^. The HCC organoid is a rectangular tissue block at day 6, with a length of 6.5 mm, width 3.25 mm, and height 100 μm (Figure 1a). Immunofluorescence staining showed highly positive staining signals with liver cancer cells (AFP), iPSC-EC (hCD31), and iPSC-MC (vimentin) in vitro (Appendix A). We found that the co-cultured cells started to self-organize 24 h after culturing (Figure 1b). Instead of using normal EC, derived from human iPSC, KO-EC labeled with red fluorescence was used. Once the iPSC-MC are added, they will be more easily organized into 3D HCC organoids (Appendix A). The functional activity of the HCC organoid was assessed, by measuring ALB production. ELISA assay of the ALB protein was performed, to measure the ALB secreted by the HCC organoid. The ELISA assay results show that the amount of ALB secretion in the HCC organoid was significantly increased from day 2 to day 4, and from day 4 to day 6 (Figure 1c). The continuous increase in ALB indicates that the liver cancer cells (Huh7-Luci) in the HCC organoid could have functioned normally. The best ratio of Huh7-Luci/iPSC-EC/iPSC-MC in the Ibidi culture-insert 2 well system was examined by measuring the luciferase signal of Huh7-Luci cells at different culture ratios. A higher luciferase signal indicates the presence of many liver cancer cells (Huh7-Luci). Our results showed that the culture consisted of HCC organoids (Huh7-Luci/iPSC-EC/iPSC-MC), with a ratio of 10:2:2 having the highest luciferase signal, indicating that this is the best cell ratio for HCC organoid formation (Figure 1d).

### 3.2. Implantation of HCC Organoid Onto the Mouse Liver Surface

After successful generation of the HCC organoid in vitro, we further evaluated the function of this HCC organoid, by implanting it to our mouse model. UH was used to disrupt cells and tissues through ultrasonic waves and cavitation. After disrupting the surface of the median lobe of the mouse liver, we implanted the HCC organoid into the damaged area, followed by capping with UPAL gel on the implanted HCC organoid (Figure 2a). In our previous study, an 18G needle was used to disrupt the liver tissue [33]. However, in the current study, we replaced the needle with UH, because using an 18G needle might increase bleeding and reduce the post-operative survival rate. First, we compared the potential effect of UPAL gel capping on HCC organoids on mouse survival rate. After implantation, the HCC organoid was either treated with UPAL gel capping (+UPAL) or without UPAL gel capping. There were no significant differences in the survival rate between the +UPAL and without UPAL groups (Figure 2b). Liver cancer was developed in all six mice in the +UPAL group, but in only four out of six mice in the group without UPAL. The lower liver cancer development rate in the without UPAL group was also found in the repeated experiment (engraftment rate in +UPAL group: 100%, engraftment rate in without UPAL group: 66.7%). Next, we compared the tumor size between the +UPAL and without UPAL groups, by in vivo BLI (Figure 2c). Overall, after the HCC organoid was attached to the mouse liver surface successfully, there was no significant difference in tumor size between the +UPAL and without UPAL groups (Figure 2c). The liver cancer cells were both observed by implanting with HCC organoid capping with UPAL, or not. The morphology between the +UPAL and without UPAL groups was compared (Figure 2d). The HE staining method was applied on liver sections from the +UPAL and without UPAL groups, the results revealed that both groups have a similar histology; the liver cancer cells (green arrows) are well differentiated and interdigitate with normal hepatocytes. We have found liver cancer cells with irregular nuclear contours (Figure 2e). The immunofluorescence staining showed that highly positive staining signals with liver cancer cells (AFP), and proliferate cell (Ki67) and MC (vimentin). The positive rate of the MC area in the HCC organoid was 9.2%. However, iPSC-EC (hCD31)-positive signals were not detected (Appendix A).

### 3.3. Generation of Liver Cancer Mouse Model after Different Liver Cancer Organoid Implantation In Vivo

The Huh7-Luci, Huh7-Luci + EC (Huh7-Luci co-cultured with iPSC-EC), Huh7-Luci + MC (Huh7-Luci co-cultured with iPSC-MC), Huh7-Luci + EC + MC (Huh7-Luci co-cultured with iPSC-EC and iPSC-MC), required six days to form a liver cancer organoid, which can be used for implantation. The secretion of ALB of the HCC organoid, from Huh7-Luci, Huh7-Luci + EC, Huh7-Luci + MC, Huh7-Luci + EC + MC at day 6, has been tested. Huh7-Luci + MC and Huh7-Luci + EC + MC significantly increased ALB secretion at day 6, as compared to the other two groups (*p* < 0.0001). However, there were no statistical differences in ALB secretion between the organoid of Huh7-Luci and Huh7-Luci + EC (*p* > 0.05) (Figure 3a). It has been indicated that MC derived from human iPSC promotes the formation of HCC organoids in vitro. Bioluminescence results on the Huh7-Luci, Huh7-Luci + EC, Huh7-Luci + MC, and Huh7-Luci + EC + MC groups showed that the intensities of luciferase within Huh7-Luci + MC, Huh7-Luci + EC + MC significantly increased at day 6, as compared to the other two groups (*p* < 0.05) (Figure 3b). The BLI of luciferase activity in the implanted Huh7-Luci + EC, Huh7-Luci + MC, and Huh7-Luci + EC + MC show more bioluminescent signal than the Huh7-Luci group, *p* < 0.05. The addition of iPSC-EC and iPSC-MC promoted liver cancer in vivo, especially iPSC-MC (Figure 3c). The images of the liver have shown four groups. We have observed liver cancer in all the groups. However, the tumors in the Huh7-Luci + EC + MC group are the largest (Figure 3d). The HE staining of the liver slides of the four groups revealed a similar histology; the liver cancer cells interdigitate with normal hepatocytes. The histological section of the liver can observe liver cancer cells (green arrows) and iPSC-MC (yellow arrows), but iPSC-EC were not detected (Figure 3e).

### 3.4. Implantation of HCC Organoid to the Liver with Fibrosis TME

To study HCC tumor development under the liver fibrosis TME, we orthotopically implanted HCC organoids (HO) into NOD-SCID mice, and induced liver fibrosis in the mice. The liver with a fibrosis TME was induced, by injecting TAA into the mice, starting at 1-week post-implantation. TAA injection was conducted three times per week, for four consecutive weeks. The results of the in vivo BLI showed that the tumor size in the HO-implanted mice that received TAA injection (HO + TAA) is not significantly different compared to the HO-only groups (2 weeks and 3 weeks after TAA injection) (Figure 4a). The tumor size in the HO + TAA group was significantly larger than the HO-only group at 4 weeks after the TAA injection, *p* < 0.0001, indicating that the TAA induced liver fibrosis TME in the animal model promotes HCC tumor growth (Figure 4a). Dissection of the HO + TAA mouse after 4 weeks after the TAA injection, further confirmed that the region of liver cancer in TAA-injected mice is larger than the HO implanted-only group (Figure 4b). The HE staining results of the liver section also reveal that the liver tissue in the TAA injection group shows portal–central fibrotic septa and nodule formation. However, it cannot be observed in the HO group (Figure 4c). In contrast, the histological structure of the tumor tissue was not different between these two groups. Sirius red staining showed that after 4 weeks of TAA injection, high fibrosis levels in the liver lobules and cellular structures could be observed in the TAA injection groups (Figure 4c). The fibrosis region was 5.5% of the total area in only the TAA injection 4 weeks group and 5.4% in the HO + TAA injection 4 weeks group. There are no significant differences in fibrosis area between these two groups (Figure 4d). It seems that the fibrosis TME could promote tumor proliferation, while tumor proliferation cannot promote the fibrosis level in the fibrosis liver in our experimental result. This promotion process seems to be one-way. The survival rate is not significantly different between the HO and HO + TAA injection 4 weeks groups (Figure 4e). The mice that received HO implantation, followed with 4 week TAA injection, had a lower body weight than the HO group and TAA injection-only group, *p* < 0.05 (Figure 4f).

### 3.5. Implantation of HCC Organoid to the Liver with NAFLD TME

To study HCC tumor development under NAFLD TME, we orthotopically implanted HO into NOD-SCID mice and induced the mice to develop NAFLD. After 1 week of HCC organoid implantation, the NOD-SCID mice were fed with CDAHFD for 5 weeks, to induce NAFLD development. Next, we monitored the tumor growth in vivo, by BLI. The BLI results showed that the tumor size in the mice fed with CDAHFD (HO + CDAHFD 5 weeks) is not significantly different compared to the mice fed with normal chow (NC) (HO + NC 5 weeks) (Figure 5a). The image of the liver at the dissection showed significant color changes in the CDAHFD group and HO + CDAHFD group (brownish white), which may affect lipid accumulation (Figure 5b). The HE staining of the liver section revealed that the liver tissue of mice fed with CDAHFD for 5 weeks is different from the HO fed with NC liver. Ballooning degeneration of hepatocytes and developed severe steatosis could be observed in the CDAHFD group and HO + CDAHFD group, by HE staining. However, the histological structure of the tumor tissue was not significantly different between these two groups (Figure 5c). Sirius red staining showed that, after 5 weeks of being CDAHFD fed, no fibrosis could be seen in the CDAHFD groups (Figure 5c). The ORO staining of the liver section showed that the liver tissue in CDAHFD 5 weeks is different from the HO-implanted liver (HO+ NC 5 weeks); several lipid droplets could be observed in the liver section in the CDAHFD group and HO + CDAHFD group (Figure 5d). There is no difference in survival rate between mice in the HO + CDAHFD 5 weeks and HO + NC 5 weeks group (Figure 5e). The mice that received HO implantation, followed by 5 weeks of CDAHFD, have a lower body weight than the HO + NC group and CDAHFD-only fed group, *p* < 0.05 (Figure 5f). The amount of TG in the liver was increased in the CDAHFD 5 weeks group and HO + CDAHFD 5 weeks group, compared to the HO + NC 5 weeks group, *p* < 0.0001 (Figure 5g).

### 3.6. Implantation of HO in the Humanized Mice

HCC was induced in the NOG and humanized mice. The images of the liver at dissection showed that the HCC of NOG mice could be observed within 2 weeks; however, HCC was hardly observed in the humanized group for 2 weeks. Moreover, compared to the NOG group at 3 weeks, the HCC was smaller in the humanized mice group (Figure 6a). BLI detected the luciferase signal of implanted HO in NOG mice 2 weeks after implantation. However, we did not detect any luciferase signal in the humanized mice group. NOG group showed significantly increased liver tumor intensities than humanized mice, after 3 weeks implantation, *p* < 0.05 (Figure 6b).

## 4. Discussion

In the previous study, we successfully generated mini-spheroid in vitro, using the hepatic endoderm, EC, and MC derived from human iPSC [30,32]. In this study, we improved the method of generating a functional sheet-like HCC organoid that can be used for implantation. This work was consistent with our previous research, wherein the HCC organoid can be obtained within one week in vitro. The difference was that the formation of the vascular endothelial network was not found in our experimental results. If the liver cancer cells are co-cultured with iPSC-EC and iPSC-MC, to investigate tumor angiogenesis, it seems that this method has a limitation in vitro. However, our control group, using hepatic endoderm, KO-EC, and iPSC-MC, had obvious vascular endothelial network formation (Appendix A). In the same co-culture device, although no vascular endothelial network was observed between iPSC-EC, iPSC-MC, and cancer cells, we speculate that the formation of the vascular endothelial network in cancer cells might be induced by the release of exosomes or paracrine factors, rather than by direct contact with cancer cells. Exosomes, membrane-bound extracellular vesicles, can modulate the TME, promoting tumor progression and angiogenesis [38]. Paracrine factors, such as angiogenic factors secreted by cancer cells, regulate the proliferation and survival of EC [39]. Some researchers have reported that the growth of EC in the 3D system requires VEGF secreted by cancer cells [40,41]. Future research is necessary to analyze the concentration of VEGF and other growth factors in ECs that are co-cultured with the cancer cells system. Whether cancer cells can inhibit the formation of the vascular endothelial network is currently unknown, and we are continuing to explore this mechanism.

The traditional method to generate a xenograft HCC animal model, involving injecting tumor cells into the animal, has the following limitations: poor engraftment rates, uncontrolled tumor size, off-target tumor development, and high risk of thromboembolism [42]. Our novel method that was used to generate a xenograft HCC animal model, overcame these limitations. It has a high engraftment rate, controlled tumor size, and the original hepatic microenvironment. UH, with which we mimic the Cavitron ultrasonic surgical aspirator (CUSA) [43], was applied in the current study, to create a wound on the surface of the liver before implantation. The use of UH has significant advantages, as it could selectively remove liver cells through high-frequency oscillating titanium tips, while retaining the bile ducts and blood vessels of the liver [44,45,46], reducing mouse mortality. If UH was not used to form a wound over the liver surface, it was difficult to see a luciferase-positive signal on the mouse after HCC organoid implantation for two weeks (Appendix A). This observation suggested that the high efficiency of engraftment could be improved by the dissociation of serosa before implantation. We predict that the wound over the liver surface, made by UH, could promote the exchange of oxygen and nutrients between the host wound surface and the implanted HCC organoid. UH, as a method producing high engraftment efficiency and low invasiveness, can be used to make liver cancer models, and has some promise for various implantation treatments.

Many studies have shown that the implanted cells or tissues often show a low engraftment rate in the recipient animal. In our previous work, we found that the UPAL gel increased the engraftment rate [33]. This time, our research also confirmed that UPAL gel could increase the engraftment rate. In the +UPAL group, the production of the HCC mice model was 100% successful, while the rate in the without UPAL group was 66.7%. The high engraftment rate in the +UPAL group might be because the UPAL gel helps the implanted organoids to anchor to the liver surface and prevents them from moving to other parts of the abdominal cavity. We found that after implanting fetal liver tissue onto the liver in rats with cirrhosis, the UPAL gel also increased the engraftment area compared to other capping agents [33]. However, we did not observe that the engraftment area had a significant difference between in the without UPAL and +UPAL groups this time. We do not know why the UPAL gel increased the engraftment rate in the current study, but had no effect on the engraftment area. The mechanism of the effects of UPAL gel in the engraftment area is still unclear. Further research into the mechanism of UPAL gel might be important in future work. Some reports have shown that UPAL gel can suppress interleukin-6 (IL-6) production in rabbit discectomy models [47]. IL-6 is generally known to be one of the major critical cytokines in the TME, promoting tumor proliferation and differentiation [48,49]. IL-6 can facilitate tumorigenesis by regulating tumor cell surface markers and various signaling pathways (such as metabolism [50,51], apoptosis [52,53], survival [54,55], proliferation [56,57], angiogenesis [58], and invasion [59,60]). We have not seen tumor proliferation or reduction. We speculate that UPAL gel affects some cytokines, such as the suppression of IL-6 secretion in the TME, which inhibits the growth of cancer cells, while increasing the engraftment rate of implanted cells. We are still studying the mechanism of the interaction between the HCC organoid and UPAL gel.

Our novel HCC animal model is a useful tool with which to study the mechanism of the TME in HCC tumor development. When the HCC organoid is generated in vitro, the TME could be manipulated by adjusting the iPSC-EC and iPSC-MC composition. It was found that the addition of iPSC-MC alone to the Huh7-Luci cells could significantly promote ALB secretion, suggesting that MC might promote the growth of Huh7-Luci. After implantation, the tumors that were developed from the HO generated through the co-culturing of Huh7-Luci, iPSC-EC, and iPSC-MC had the largest tumor volume. This result indicated that both iPSC-EC and iPSC-MC are important in promoting HCC tumor growth. We reasoned that iPSC-EC might promote HCC tumor growth through angiogenesis. Tumor proliferation and migration are generally known to depend on angiogenesis [61]. While angiogenesis is formed after EC proliferation and migration [62,63]. When EC was added and co-cultured with Huh7-Luci in vitro, there was no significant increase in the tumor cell number (Huh7-Luci). However, the tumor size, after implantation in vivo, showed a significant difference. Although the volume of HCC organoid increased significantly after adding EC derived from iPSC, our immunofluorescence test failed to detect hCD31, a human EC marker gene, indicating that the growth of the HCC tumor is not due to an increase in iPSC-EC number. We speculated that the iPSC-EC might migrate from the tumor after implantation. Therefore, we further speculate that iPSC-EC might promote tumor growth through distal paracrine secretion, instead of inducing tumor cells growth through direct proximal cell contact. Several reports have shown that TGF-β, a paracrine factor that is secreted by EC, can stimulate the proliferation and metastasis of HCC via epithelial-to-mesenchymal transition [64]. This finding suggested how iPSC-EC in the HCC TME promotes HCC tumor growth. Although the vascular endothelial network formation was not found in vitro, our histological and BLI analyses showed that the lack of a vascular endothelial network does not affect the tumor growth and functions.

Many methods to develop animal models have been developed; however, none of these methods can replicate the liver pathological microenvironment. To simulate the development of clinical HCC, we established a reliable model of HCC in the mouse, using different cancer organoid technology, and we examined the effects of different host liver microenvironments, such as NAFLD fibrosis on liver cancer development. Advanced liver fibrosis and cirrhosis are the main risk factors for liver cancer. In the background of liver cirrhosis, as many as 90% of patients eventually become HCC [65,66]. In our study, HCC lost contact with normal cells or tissues, and lost the normal TME or extracellular matrix, which may further promote the development of HCC. However, if the degree of fibrosis was not particularly large, this phenomenon of promoting the development of HCC was not obvious. NAFLD is also considered to be another important factor in promoting the growth of HCC [67]. After the mice ate CDAHFD food for five weeks, although the TG in the liver tissue increased, the severity of liver fibrosis was not obvious compared with TAA. It seems that a high degree of fibrosis promotes the development of HCC. Some other clinical studies have reported that liver stiffness and high fibrosis indicators are positively related to liver cancer [68,69].The fibrotic microenvironment secretes many inflammatory cytokines, leading to the production of reactive oxygen species (ROS). Chronic ROS exposure lays the foundation for the development of HCC. The mechanisms are related to the ROS-mediated DNA damage [70]. Reducing the production of ROS will suppress the development of liver cancer [71,72]. Our experimental results may also provide clinical inspiration. In the treatment of HCC patients, in addition to inhibiting tumor growth, it is also necessary to pay attention to the progress of liver fibrosis.

Many researchers use immunodeficient mice as animal models for pre-cancer research. However, immune cells play an important role in cancer development and treatment. The humanized mouse, immunodeficient mice with human immune systems, was successfully made, using CD34 + cells from umbilical cord blood donors [73,74,75]. In the engraftment of the 100% HCC organoid, the luciferase-positive signals were observed two weeks after implantation in the NOG group. However, the humanized mice did not produce luciferase-positive signals until three weeks after HCC implantation. Compared to NOG mice, implanting HO for mouse humanization has resulted in delayed engraftment. In our experiment, we only observed the phenomenon. However, we do not know the specific mechanism. Immune monitoring is closely related to the occurrence of tumors [76]. Our follow-up experiments will focus on the relation between tumors and immunity, such as how tumor cells evade immune surveillance three weeks after implanting the HCC organoid. Further studies are also needed in order to understand the way in which antitumor immune responses are generated in humanized mice, by analyzing various immune cells, cytokines, and the TME. In addition to immunity factors, non-immune factors, such as the TME, may also affect the human tumorigenesis process in mouse models.

This new orthotopic implantation method is not only suitable for one liver cancer cell line (Huh7), other liver cancer cell lines, HepG2, can also be made into a liver cancer model successfully (Appendix A). In addition, to prove that the implantation method that was developed in the current study can be applied to generate an orthotopic cancer animal model other than HCC, we also successfully generated a highly metastatic pancreatic cancer model, by using the same orthotopic implantation method. After orthotopic surgical implantation for 8 weeks, pancreas cancer could be observed in the pancreas, and distant metastasis liver lesions were also found. Liver metastasis was found in two out of four mice. The HE staining of the pancreas and liver slides revealed that the cancer cells could interdigitate with normal pancreatic cells and hepatocytes (Appendix A). These results confirm that this implantation method’s ability is suitable for making liver cancer and pancreatic cancer models.

Recently, several studies have reported the application of organoid implantation in cancer research, and have shown it to be very useful in the investigation of TME mechanisms and cell dynamics in tumors [29,77]. For example, colorectal cancer organoids containing collagen have been grafted over the sub-mucosal layer of the cecal wall, to induce colorectal cancer in mice [29]. However, organoid implantations apply biomimicry scaffold materials to the cecal wall, leaving no direct contact between the implanted organoid and recipient tissue. As a result, this approach may cause the tumor to be limited to the surface of the grafted organ or tissue [29]. This situation may limit the ability of organoid implantation to closely resemble tumor development in patients, wherein tumor growth is in all directions within the organ or tissue. In the current study, we applied UPAL to the implantation site and grafted the organoid onto it. The UPAL acts as a glue-like substance, to support and connect the implanted organoid and recipient liver tissue. As a result, the implanted tumor organoid not only grew on the surface of the organ, but in all directions. In addition, most of the reported organoid implantations use a mixture of organoid and derived biomimicry scaffold materials, such as Matrigel extracted from animals [77]. These materials may carry animal virus, which might affect the tumor growth and data accuracy. In our work, we used UPAL gel derived from the plant polysaccharide matrix, extracted from the cell wall of brown algae, which may not carry animal pathogens [78]. Our study showed that there was no significant difference in tumor size with and without UPAL. In addition, we observed that liver cancer could be induced in humanized mice. We decided on UPAL as a capping agent, because it not only fixes the HCC organoid, but it is also a proven safe clinical injury recovery dressing material that can provide a moist wound environment to facilitate healing [79,80,81]. With our method, we could create an orthotopic implantation mouse model surrounding the human-like TME, which will help us to monitor cell dynamic development, tumorigenesis, and metastasis in the future.

In this study, we introduced a new method of producing tumors, and observed the effects of the TME and immunity on tumor formation. However, we did not show further mechanisms, such as the kind of cytokines that are released after the addition of iPSC-EC, iPSC-MC, or a fibrotic microenvironment, or the signal pathways related to the liver cancer cell growth that is activated by these released factors. A better understanding of the interaction between cancers and the TME may help in identifying new ways to treat HCC.

## 5. Conclusions

In conclusion, we have developed a novel, robust, effective orthotopic implantation method, and have applied this method to generate an HCC animal model. The major advantage of this animal model is that we can manipulate and mimic the TME factors of HCC tumors, providing an excellent tool to study the roles of the TME in tumor development. Moreover, we also successfully generated a highly metastatic pancreatic cancer model using the same orthotopic implantation method. It showed that this implantation method could be applied to generate other cancer animal models as well. Our method is particularly suitable to produce parenchymatous organ cancer models, considering the use of UH.

## Figures and Tables

**Figure 1 cancers-13-03997-f001:**
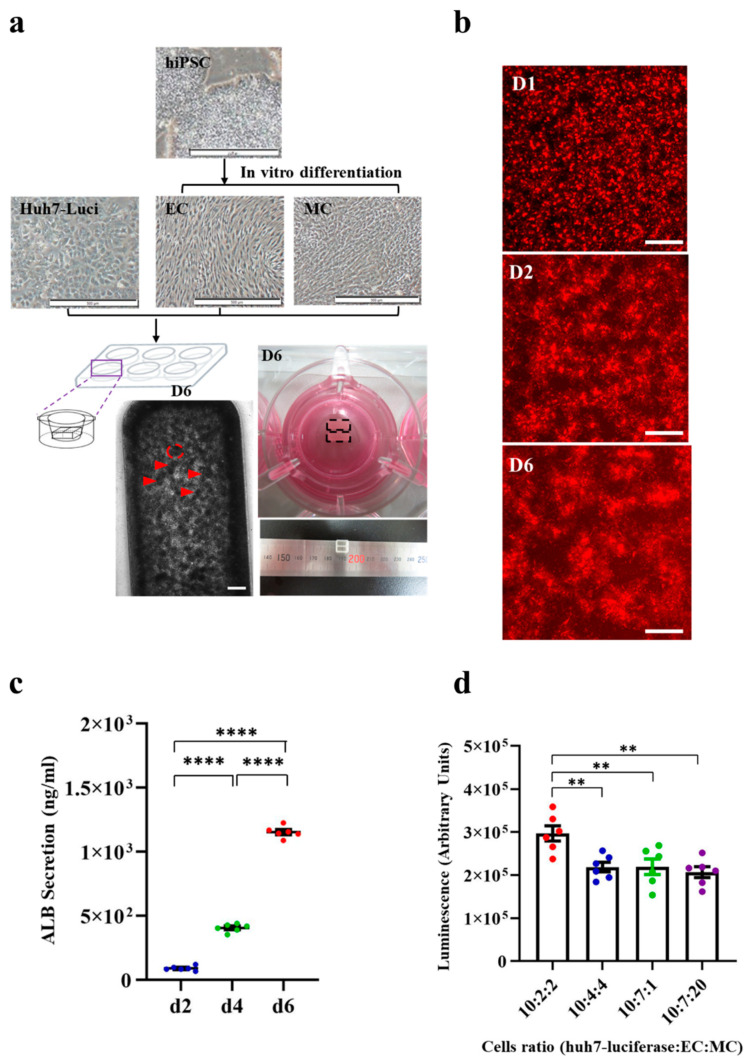
Generation of human hepatocellular carcinoma (HCC) organoid. (**a**) Bright-field image of the protocol for the generation of human HCC organoid. Morphology of human HCC organoid in the Ibidi culture-insert 2 well system at day 6. After 6 days of cultivation, the human HCC organoid was generated. Red arrows indicate single mini HCC organoid. The red dotted line indicates the area of each mini HCC organoid. The black dotted line indicates the rectangular tissue block. The scale bar represents 500 μm. (**b**) Live imaging of KO-EC with Huh7-Luci and iPSC-MC in the Ibidi culture-insert 2 well system at day 1, 2, and 6. The scale bar represents 200 μm. (**c**) ELISA of ALB secretion in HCC organoid at day 2, 4, and 6. One-way ANOVA followed by Tukey’s multiple comparisons test (*n* = 6/group). (**d**) Luciferase measurement of Huh7-Luci in HCC organoids generated by different composition ratio of Huh7-Luci, iPSC-EC, and iPSC-MC. One-way ANOVA followed by Tukey’s multiple comparisons test (*n* = 6/group). ** *p* < 0.01, **** *p* < 0.0001.

**Figure 2 cancers-13-03997-f002:**
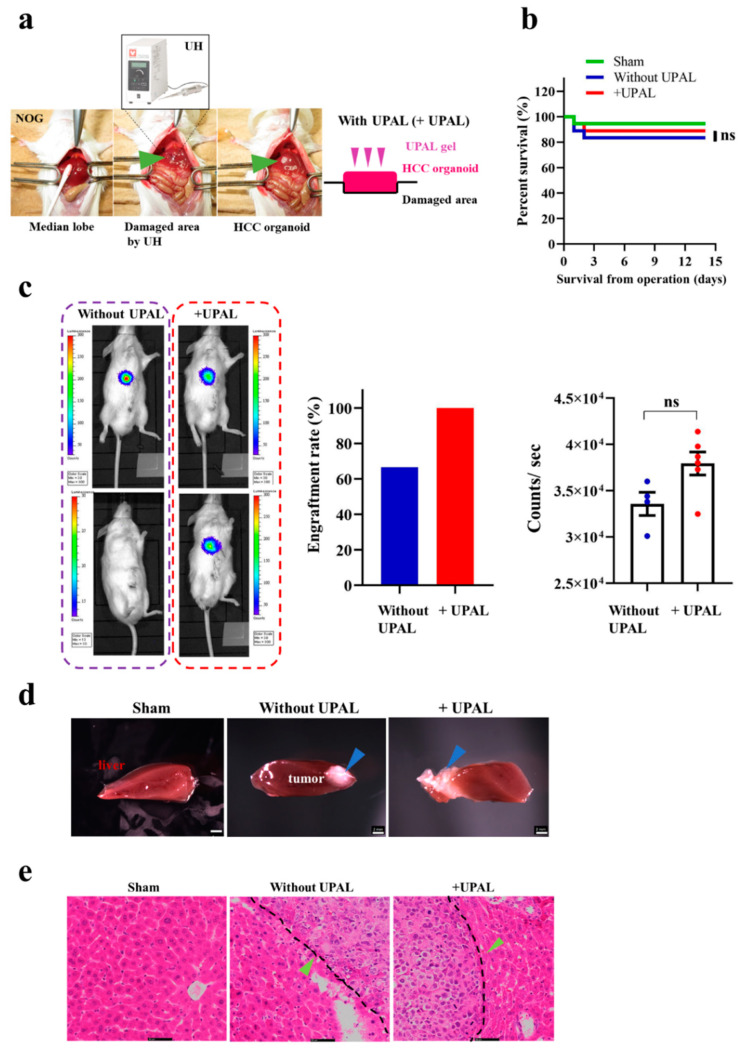
Implantation methods of HCC organoid onto the liver surface. (**a**) Implantation procedure of HCC organoid into mouse liver. (**b**) The survival rate of ultra-purified alginate gel-capped (+UPAL) HCC organoid implantation. Without UPAL gel-capped HCC implanted group and Sham were used as control. Log-rank test (*n* = 18). (**c**) Non-invasive BLI of mouse implanted with HCC organoid; without UPAL gel-capped group (left) and UPAL-capped gel (+UPAL) group (right). Acquisitions of the fluorescence emission were performed at 1 min at a binning of 100 pixels. The relative light units/pixel is indicated in the color scale bar. Mann–Whitney *U* test (*n* = 6/group). (**d**) Morphology analysis of mouse liver tissues implanted with HCC organoid; sham (left), HCC organoid without UPAL gel capping (center), and with UPAL gel capping (right). Blue arrows indicate tumor tissue. The scale bar represents 2 mm. (**e**) Hematoxylin and eosin (HE) staining of mouse liver after 2 weeks of HCC organoid implantation. The black dotted line indicates the junction of normal liver tissue and tumor tissue. Green arrows indicate tumor cells. The scale bar represents 50 μm. ns: not significant.

**Figure 3 cancers-13-03997-f003:**
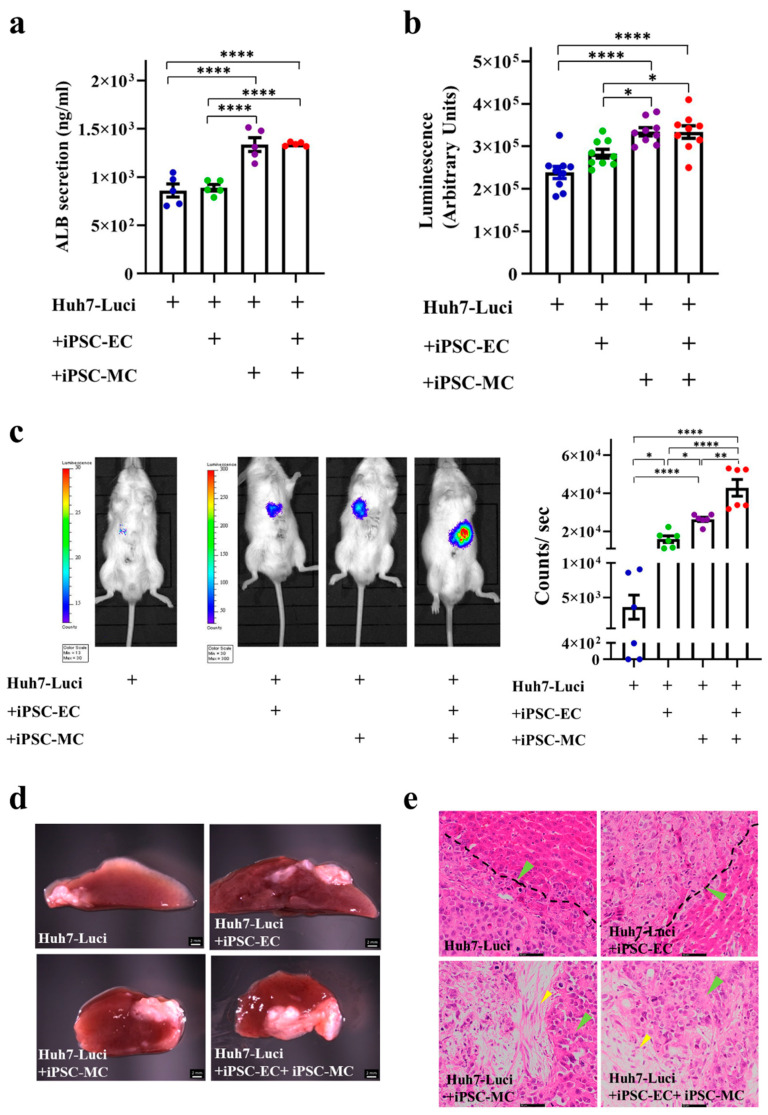
Liver cancer organoid can separate into four groups in vitro and in vivo. (**a**) Albumin (ALB) secretion of Huh7-Luci, Huh7-Luci + iPSC-EC, Huh7-Luci + iPSC-MC, and Huh7-Luci + iPSC-EC + iPSC-MC organoid at day 6. One-way ANOVA followed by Tukey’s multiple comparisons test (*n* = 5/group). (**b**) Bioluminescence intensities of luciferase activity in Huh7-Luci, Huh7-Luci + iPSC-EC, Huh7-Luci + iPSC-MC, Huh7-Luci + iPSC-EC + iPSC-MC group at day 6 in vitro. One-way ANOVA followed by Tukey’s multiple comparisons test (*n* = 9/group). (**c**) BLI of luciferase activity in mice implanted with different liver cancer organoid groups in vivo. Left to right: Huh7-Luci, Huh7-Luci + iPSC-EC, Huh7-Luci + iPSC-MC, Huh7-Luci + iPSC-EC + iPSC-MC show pseudo color images of bioluminescent signals from the liver. Acquisition times were set as 1 min at a binning of 100 pixels. Relative light units/pixels are indicated in the color scale bar. One-way ANOVA followed by Tukey’s multiple comparisons test (*n* = 6/group). (**d**) Morphology analysis of organoids composed of Huh7-Luci, Huh7-Luci + iPSC-EC, Huh7-Luci + iPSC-MC, Huh7-Luci + iPSC-EC + iPSC-MC that were implanted onto the liver surface, followed by UPAL gel capping. The scale bar represents 2 mm. (**e**) HE staining of mice liver after 2 weeks of different liver cancer organoid implantation. The black dotted line indicates the junction of normal liver tissue and tumor tissue. Green arrows indicate tumor cells; yellow arrows indicate iPSC-MC. The scale bar represents 50 μm. * *p* < 0.05, ** *p* < 0.01, **** *p* < 0.0001.

**Figure 4 cancers-13-03997-f004:**
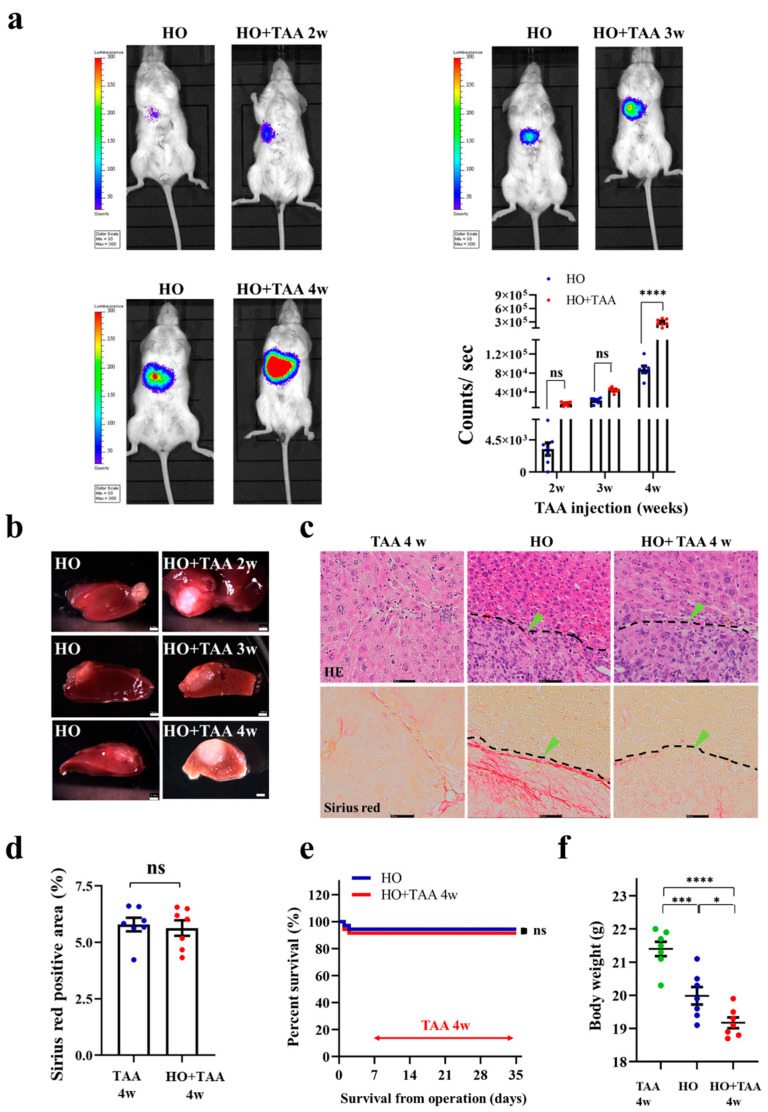
Implantation of HCC organoid into the liver with fibrosis TME. (**a**) BLI of luciferase signal in mice with or without fibrosis TME. Left to the right column (top): implanted HCC organoid only (HO), implanted HO + TAA injection 2 weeks. Third to the fourth left panel: implanted HO only, implanted HO + TAA injection 3 weeks. Left to the right column (bottom): implanted HO only, implanted HO + TAA injection 4 weeks (implanted HO only was regarded as control group at the same time); tumor cells were illuminated as pseudo color. Two-way ANOVA test, (*n* = 7/group). (**b**) Images of implanted HO without TAA injection at a different times (left panel), and implanted HO + TAA injection 2 weeks, 3 weeks, 4 weeks (right panel); the scale bar represents 2 mm. (**c**) HE staining and Sirius red staining of the liver treated with TAA injection 4 weeks only, HO and HO + TAA injection 4 weeks (left to right). In the HE staining results (top), the black dotted line indicates liver tissue and tumor tissue junction. Green arrows indicate tumor cells. Scale bar represents 50 μm. (**d**) Measurement of the fibrosis area in TAA injection 4 weeks group and HO + TAA injection 4 weeks group was performed with Mann–Whitney *U* test, (*n* = 7/group). (**e**) The survival rate of HO and HO + TAA injection 4 weeks. Log-rank test (*n* = 14). (**f**) Body weight of mice. One-way ANOVA followed by Tukey’s multiple comparisons test (*n* = 7/group). * *p* < 0.05, *** *p* < 0.001, **** *p* < 0.0001; ns: not significant.

**Figure 5 cancers-13-03997-f005:**
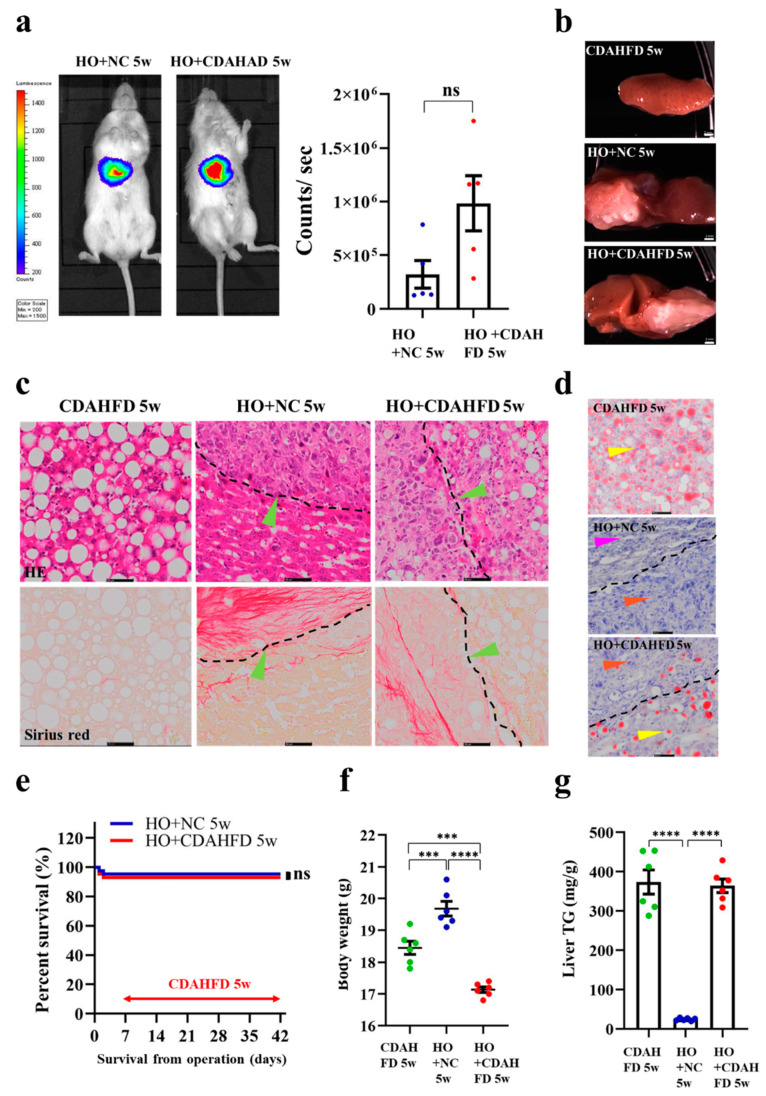
Implantation of HCC organoid to the liver with NAFLD TME. (**a**) Bioluminescence imaging that shows luciferase signal in mice fed with or without CDAHFD. Bioluminescence signal is illustrated in pseudo color. Mann–Whitney *U* test (*n* = 5/group). (**b**) Images of liver tissues. Top to bottom: liver tissue from mice fed with CDAHFD 5 weeks only, liver tissues from mice implanted with HCC organoid fed with NC (HO + NC 5 weeks), liver tissues from mice implanted with HCC organoid and fed with CDAHFD (HO + CDAHFD 5 weeks). Scale bar represents 2 mm. (**c**) HE staining and Sirius staining of mice liver in the following different groups: fed CDAHFD 5 weeks only (left top panel), HO + NC 5 weeks (middle top panel), HO + CDAHFD 5 weeks (right top panel). The black dotted line indicates the junction of liver tissue and tumor tissue. The bottom panel shows the Sirius red staining result. Green arrows indicate tumor cells. The scale bar represents 50 μm. (**d**) The figure shows the ORO staining of mice liver in the following different groups: fed CDAHFD 5 weeks only (top), implanted HO + NC 5 weeks (middle), implanted HO + fed CDAHFD 5 weeks (bottom). The black dotted line indicates the junction of liver tissue and tumor tissue. Orange arrows indicate tumor cells, yellow arrows indicate NAFLD cells, purple arrows indicate normal liver cells. The scale bar represents 50 μm. (**e**) The survival rate of HO + NC 5 weeks and HO + CDAHFD 5 weeks. Log-rank test, *n* = 12. (**f**) Body weight of the mice. One-way ANOVA followed by Tukey’s multiple comparisons test (*n* = 6/group). (**g**) Liver TG. One-way ANOVA followed by Tukey’s multiple comparisons test (*n* = 6/group). *** *p* < 0.001, **** *p* < 0.0001; ns: not significant.

**Figure 6 cancers-13-03997-f006:**
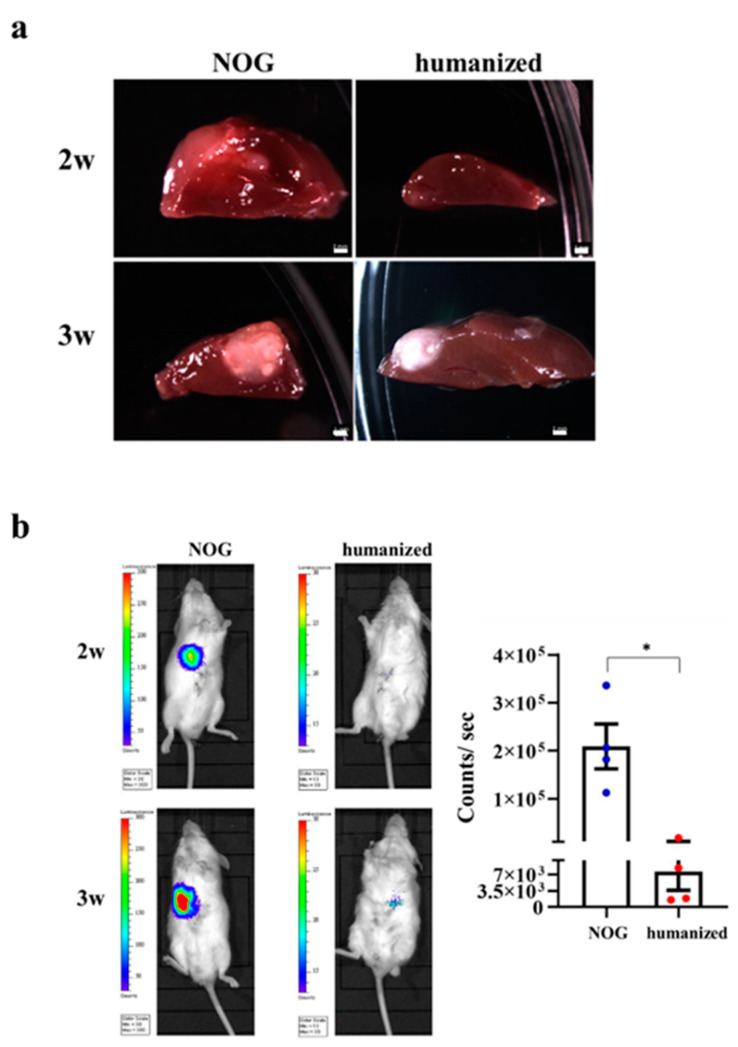
HCC organoid implantation in the humanized mice. (**a**) Morphology analysis on liver tissues from NOG and humanized mice. NOG mice (left), humanized mice (right). The top panel is the result of HO implantation 2 weeks. The bottom panel is the result of HO implantation 3 weeks. Scale bar represents 2 mm. (**b**) BLI of luciferase signal in humanized and NOG mice. Bioluminescence signal is illustrated in pseudo color. Four panels (first top panel: implanted HO into NOG mouse at 2 weeks, second top panel: implanted HO into humanized mouse at 2 weeks, bottom panel means after implantation in the different mouse at 3 weeks. Mann–Whitney *U* test (*n* = 4/group). * *p* < 0.05.

## Data Availability

Data that are presented in this study are available on request from the corresponding author.

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
