# Peer review of "A Novel Orthotopic Liver Cancer Model for Creating a Human-like Tumor Microenvironment"

_cancers, 2021, doi:10.3390/cancers13163997_

Round 1
Reviewer 1 Report
This excellent and innovative research from Qiu et al includes important advances in the field of models of HCC [Hepatocellular carcinoma] that better recapitulates human disease.
The most important components of an ideal model for TME [tumor microenvironment] are 1. Human cancer cells, 2. Human fibroblasts, 3. Human leucocytes. This new model from Qiu et al has achieved two of these 3 components and reports important information about the third.
The demonstration that 4 weeks of TAA increases HCC growth also fits with good modelling of human disease.
The level of innovation, such as abrading the liver and overlaying the HCC with alginate, is impressive.
My suggestions for improvement are:
1. Title: I do not believe that there is justification for including the word, ‘transplantation’, in the title. Because ‘transplantation’ implies transfer of a tumour from one animal or human to another, and this was not done. The HCC was implanted rather than transplanted.
- Title. Please reconsider the title so that it conveys the key message of the paper. Perhaps focus on the achievement of an orthotopic model that includes Human cancer cells and Human fibroblasts in a mouse liver such that a human-like TME is partially created.
- Please comment on why the liver insults , TAA or diet, that were designed to provide a tumour-friendly disrupted liver environment were not commenced until a week after the HCC was applied to the liver. Would it not be a superior strategy to create fibrosis before introducing HCC cells to the liver?
- Choline deficient diets are not as close to human situation as is a high fat high sucrose diet , maybe with extra cholesterol. Perhaps a high fat high sucrose diet eaten for 6 weeks prior to implanting HCC would increase tumour growth. I invite comment.
- Markers of human fibroblasts or CAFs, such as FAP-alpha or FSP immunostains would be helpful to prove which stroma is human. For example, the F19 monoclonal antibody to human FAP used on frozen sections is human-specific [ W. J. Rettig, etal 1993 Regulation and heteromeric structure of the fibroblast activation protein in normal and transformed cells of mesenchymal and neuroectodermal origin. Cancer Research, 53(14): 3327-35. ].
- Fig S2 needs to provide also the single-colour images.
- Fig S2 shows a small area of liver with vimentin stain. This does not tell the reader the scope or extent of human fibroblast cell growth in the mouse liver. Please provide images of larger areas of liver and quantify it.
- A limitation is the ability to detect luciferase deep in the body. This is probably a reason why the HCC is applied to the top surface of the liver; to assist in detecting the luciferase signal. Was iRFP720 considered as an alternative label that is more readily detected in deep tissue because it emits in the extreme infra-red range.? [ A Wilson, etal 2019 Non-invasive fluorescent monitoring of ovarian cancer in an immunocompetent mouse model. Cancers, 11(1): E32. doi 10.3390/cancers11010032 PMID 30602661]
- A report some years ago claimed that mouse stroma might be humanised by exploiting the ability of an AAV [AAV7?] to enter fibroblasts and deliver a DNA cargo. Was this investigated as a means of humanising mouse stroma?
- A number of high-power photomicrographs should be added in order to show clearly the morphology of the cells in the livers.
- The discussion could have less repeating of results. E.g. p.18 line 598 and nearby.
- Page 1 line 29. Is ‘homogeniser’ the correct word to use here? Please reconsider how to accurately explain the technique.
- Methods section:
Section 2.12. Suggestion: Use the term, ‘antigen retrieval’ to explain why citrate and autoclave are used.
- English:
There are many sentances and phrases that need improved English, especially in Methods and Discussion.
Author Response
Dear editor and reviewers,
We sincerely thank the editor and all reviewers for their invaluable comments. We have responded to each of the comments and made the required corrections accordingly.
Reviewer #1
- Title: I do not believe that there is justification for including the word, ‘transplantation’, in the title. Because‘transplantation’ implies transfer of a tumour from one animal or human to another, and this was not done. The HCC was implanted rather than transplanted.
Response:
Thank you for your comment. We had changed our title and replaced the term “transplantation” with “implantation”. In addition, we also made corresponding changes in the main text and use “implantation” instead of “transplantation”.
- Please reconsider the title so that it conveys the key message of the paper. Perhaps focus on the achievement of an orthotopic model that includes Human cancer cells and Human fibroblasts in a mouse liver such that a human-like TME is partially created.
Response:
Thank you for the suggestion. We have changed the title as “A novel orthotopic liver cancer model for creating a human-like tumor microenvironment”.
- Please comment on why the liver insults , TAA or diet, that were designed to provide a tumour-friendly disrupted liver environment were not commenced until a week after the HCC was applied to the liver. Would it not be a superior strategy to create fibrosis before introducing HCC cells to the liver?
Response:
We agreed that inducing liver fibrosis in the mouse before HCC implantation might be a better strategy to simulate actual human tumour development. In fact, we did perform a preliminary experiment and tried to induce liver fibrosis/NAFLD (TAA injection and CDAHFD diet) 2 weeks prior to HCC implantation (data not shown in the manuscript). However, the survival rate of TAAï¹¢HCC/ CDAHFD ï¹¢HCC group is not high (64 %). We dissected the dead mice but failed to identify the cause of death. So far we suspected that the inflammation of the surface of a wound in liver is very strong. In the case of impaired liver function, HCC organoid aggravates the inflammatory response. However, the treatment control group (92 %) may release some inhibitory inflammatory factors. So we modified the method to induce liver fibrosis or NAFLD condition after tumour implantation.
- Choline deficient diets are not as close to human situation as is a high fat high sucrose diet, maybe with extra cholesterol. Perhaps a high fat high sucrose diet eaten for 6 weeks prior to implanting HCC would increase tumour growth. I invite
comment.
Response:
Thank you for the comment. We understand the reviewer concern about CDAHFD might not be the best option to induce fatty liver because it is not close to human diet. Before starting the current study, we performed some preliminary experiments to compare and determine which diet should be applied to induce NAFLD in mouse (data not shown in the manuscript). We have tried to induce fatty liver in mouse with 1) high-fat diet (HFD), 2) steatohepatitis-inducing high-fat diet (STHD) and 3) CDAHFD. We compared these diet and found that HDF and STHD diet need long time to cause fatty liver. Moreover, the incidence (effect) of fatty liver is not as good as CDAHFD, and the size/area of ​​fatty liver is also inconsistent. CDAHFD is a very common method and has been demonstrated to produce steatohepatitis, liver fibrosis, and liver cancer in mouse and rat to mimic human NAFLD/NASH [1,2]. Although it is not a human diet, however, no major side effects have been reported. Because research focus on the effect of HCC implantation, a rapid, robust, and controllable method is very important. In summary, I chose to use CDAHFD instead of other diet. Currently, NAFLD/NASH mouse model diet is controversial, there is still no conclusion. Our pre-experiment did not compare a high fat high sucrose diet. Thank you for your comment. In the future experiment, we will consider to add a high fat high sucrose diet group.
- Markers of human fibroblasts or CAFs, such as FAP alpha or FSP immunostains would be helpful to prove which stroma is human. For example, the F19 monoclonal antibody to human FAP used on frozen sections is human-specific [ W. J. Rettig, et al 1993 Regulation and heteromeric structure of the fibroblast activation protein in normal and transformed cells of mesenchymal and neuroectodermal origin. Cancer Research, 53(14):3327-35. ].
Response:
Thank you for your comment. In our future experiments, we will try to use two new markers which you recommend to observe the condition of human fibroblasts in the tumor microenvironment.
- Fig S2 needs to provide also the single-colour images.
Response:
We have added a single-colour images in the Supplement Fig S3 a.
- Fig S2 shows a small area of liver with vimentin stain. This does not tell the reader the scope or extent of human fibroblast cell growth in the mouse liver. Please provide images of larger areas of liver and quantify it.
Response:
We have added an image of larger liver area in the supplementary data (Figure S3 b) and quantified the vimentin-stained cell area (please see page 9, line 380).
- A limitation is the ability to detect luciferase deep in the body. This is probably a reason why the HCC is applied to the top surface of the liver; to assist in detecting the luciferase signal. Was iRFP720 considered as an alternative label that is more
readily detected in deep tissue because it emits in the extreme infra-red range.? [ A Wilson, etal 2019 Non-invasive fluorescent monitoring of ovarian cancer in an immunocompetent mouse model. Cancers, 11(1): E32. doi 10.3390/cancers11010032 PMID 30602661].
Response:
We really appreciated your suggestion of using iRFP720 [3,4] for in vivo imaging of cells inside the liver. We will consider to use it in the following studies. In our previous preliminary experiments, HCC was applied to between the median and right/left lobes of the liver (not top surface of the liver). We also can observe the positive signals. Because there is no median lobe covering like mouse in the anatomy of human liver, this is why we recommend surface implantation.
- A report some years ago claimed that mouse stroma might be humanized by exploiting the ability of an AAV [AAV7?] to enter fibroblasts and deliver a DNA cargo. Was this investigated as a means of humanizing mouse stroma?
Response:
Sharma et al. found that AAV 2/6 transduced human corneal fibroblasts have the highest efficiency and did not produce significantly cell death in vitro by comparing three different AAV vectors (AAV2/6, AAV2/8, AAV 2/9) [5]. Other researchers also reported that tyrosine-mutant AAV2 vectors have highly transduction efficiency in mouse embryonic fibroblasts [6]. Currently, Gabriel et al. have reported that the feasibility of AAV transduced mouse mesenchymal stromal cells as a vehicle for treating a murine liver injury model [7]. In our experiment, the recipient mice were irradiated with 160 cGy of X-rays to eliminate mouse hematopoietic stem cell and the umbilical cord blood CD34+ cells were implanted by intravenous injection to create the humanized mouse. At present, I have thought for humanizing immunity of attacking human cancer tissue. I really appreciate for the nice suggestion of humanizing mouse stroma by using AAV vectors. In the future experiment, we will consider to use AAV vectors for humanizing mouse stroma cells for implantation of human cancer tissue.
- A number of high-power photomicrographs should be added in order to show clearly the morphology of the cells in the livers.
Response:
We have added some high-power photomicrographs of HE, Sirius red, and Oil Red O staining in the manuscript. (Fig 2 e, page 10, line 382; Fig 3 e, page 12, line 419; Fig 4 c, page 14, line 463; Fig 5 c and 5d, page 16, line 503)
- The discussion could have less repeating of results. E.g. p.18 line 598 and nearby.
Response:
We have deleted the repeating of results. (manuscript: E.g. p.18 line 598 and nearby.
revised manuscript: page 19, line 630 to page 20, line 650).
- Page 1 line 29. Is ‘homogeniser’ the correct word to use here? Please reconsider how to accurately explain the technique.
Response:
We referred the devise as “homogenizer” because its official, commercial name is “Ultrasonic Homogenizer” by its manufacturer Yamato Scientific (https://www.yamato-scientific.com/product/category/science/stirrer/honogenizer/). We think it may confuse the readers if we refer the “Ultrasonic Homogenizer” as something else. To maintain uniformity and avoid confusion, we will maintain to use of “homogenizer” in this manuscript.
- Methods section: Section 2.12. Suggestion: Use the term, ‘antigen retrieval’ to explain why citrate and autoclave are used.
Response:
Thank you for the suggestion. We have added explanation about the use of citrate and autoclave for antigen retrieval in the Methods section 2.12. Please refer to page 6, line 250 until line 254.
- English: There are many sentances and phrases that need improved English, especially in Methods and Discussion.
Response:
Thank you for your suggestion. We reconsidered and modified English before resubmission.
Reference:
- Denda, A.; Kitayama, W.; Kishida, H.; Murata, N.; Tsutsumi, M.; Tsujiuchi, T.; Nakae, D.; Konishi, Y. Development of hepatocellular adenomas and carcinomas associated with fibrosis inC57BL/6J male mice given a choline-deficient, L-amino acid-defined diet. Jpn. J. Cancer Res. 2002, 93, 125–132, doi:10.1111/j.1349-7006.2002.tb01250.x.
- Matsumoto, M.; Hada, N.; Sakamaki, Y.; Uno, A.; Shiga, T.; Tanaka, C.; Ito, T.; Katsume, A.; Sudoh, M. An improved mouse model that rapidly develops fibrosis in non-alcoholicsteatohepatitis. Int. J. Exp. Pathol. 2013, 94, 93–103, doi:10.1111/iep.12008.
- Wilson, A.L.; Wilson, K.L.; Bilandzic, M.; Moffitt, L.R.; Makanji, M.; Gorrell, M.D.; Oehler, M.K.; Rainczuk, A.; Stephens, A.N.; Plebanski, M. Non-Invasive Fluorescent Monitoring of Ovarian Cancer in an Immunocompetent MouseModel. Cancers (Basel). 2018, 11, doi:10.3390/cancers11010032.
- Fukuda, A.; Honda, S.; Fujioka, N.; Sekiguchi, Y.; Mizuno, S.; Miwa, Y.; Sugiyama, F.; Hayashi, Y.; Nishimura, K.; Hisatake, K. Non-invasive in vivo imaging of UCP1 expression in live mice via near-infraredfluorescent protein iRFP720. PLoS One2019, 14, e0225213, doi:10.1371/journal.pone.0225213.
- Sharma, A.; Ghosh, A.; Hansen, E.T.; Newman, J.M.; Mohan, R.R. Transduction efficiency of AAV 2/6, 2/8 and 2/9 vectors for delivering genes inhuman corneal fibroblasts. Brain Res. Bull. 2010, 81, 273–278, doi:10.1016/j.brainresbull.2009.07.005.
- Li, M.; Jayandharan, G.R.; Li, B.; Ling, C.; Ma, W.; Srivastava, A.; Zhong, L. High-efficiency transduction of fibroblasts and mesenchymal stem cells bytyrosine-mutant AAV2 vectors for their potential use in cellular therapy. Hum. Gene Ther.2010, 21, 1527–1543, doi:10.1089/hum.2010.005.
- Gabriel, N.; Samuel, R.; Jayandharan, G.R. Targeted delivery of AAV-transduced mesenchymal stromal cells to hepatic tissue forex vivo gene therapy. J. Tissue Eng. Regen. Med. 2017, 11, 1354–1364, doi:10.1002/term.2034.
Reviewer 2 Report
The authors developed a new method to generate an HCC mouse model using human HCC organoids with and without human iPSC-derived endothelial and mesenchymal cells. Similarly, the same method was used to generate a highly metastatic pancreas cancer model. These models may be useful for studying the roles of tumor microenvironment in tumor development. This manuscript is very interesting and well-organized.
Although there were not yet sufficient data to show that these models actually mimic human cancer development, these models may be valuable in studying the relationship between cancer cells and tumor microenvironment.
Author Response
Reviewer #2
- The authors developed a new method to generate an HCC mouse model using human HCC organoids with and without human iPSC-derived endothelial and mesenchymal cells. Similarly, the same method was used to generate a highly metastatic pancreas cancer model. These models may be useful for studying the roles of tumor microenvironment in tumor development. This manuscript is very interesting and well-organized. Although there were not yet sufficient data to show that these models actually mimic human cancer development, these models may be valuable in studying the relationship between cancer cells and tumor microenvironment.
Response:
Thank you for the comments. We are glad to know that you agreed this model might be valuable in studying the relationship between cancer cells and tumor microenvironment. Considering the difference of anatomical structure and genetic profiles between model animal and humans, it will be very difficult to fully mimic the actual human cancer development. We think that our current organoid implantation model is a better option compared to the traditional bidimensional model like cell cultures and somehow might be better than most animal models because our model can more closely mimic the TME.
There are still a lot of follow up studies to improve this technique as well as the model, we sincerely hope that one day this model could fully mimic the actual cancer development and contribute more to cancer treatment.
Reviewer 3 Report
This paper introduces the new methods to mimic liver cancer considering the tumor microenvironment. The tumor microenvironment has been noted for cancer research because the connection with cancer cells causes cancer growth. However, the significance and role of this method should be clarified for readers’ better understanding. In addition, the reported technologies for cancer models should be introduced in detail. Taken together, major revisions should be made before paper re-submission.
- Introduction
Cancer spheroid or organoid models in vitro and in vivo combined with tumor microenvironment have already been reported. Therefore, the authors should introduce the previous study referring to the papers for readers’ better understanding. I suggest some references.
Cancers 2020, 12(10), 2754.
Tissue Eng. Part B 2010, 16, 351.
Adv. Mater. 2020, 32, 1902007
Tissue Eng. Part A, 26, 2020, 1272-1282.
Sci. Rep. 2019, 9, 292.
2. Figure 1
The details of the organoids, such as cell number, size, each cell distribution, CYP expression, should be clarified.
3. The secretion level of cytokine or chemokine should be investigated.
4. Discussion
Organoid transplantation has already been reported. Therefore, the authors should claim the study comparing the related studies.
Author Response
Dear Editors and Reviewers,
We sincerely thank the editor and all reviewers for their invaluable comments. We have responded to each of the comments and made the required corrections accordingly.
Reviewer #3
- Introduction: Cancer spheroid or organoid models in vitro and in vivo combined with tumor microenvironment have already been reported. Therefore, the authors should introduce the previous study referring to the papers for readers’better understanding. I suggest some references.
Cancers 2020, 12(10), 2754.
Tissue Eng. Part B 2010, 16, 351.
Adv. Mater. 2020, 32, 1902007
Tissue Eng. Part A, 26, 2020, 1272-1282.
Sci. Rep. 2019, 9, 292.
Response:
In response to your comment, we have included the description of the use of cancer spheroid/organoid models in tumor microenvironment studies (page 2, line 93 to page 3, line 122).
- Figure 1: The details of the organoids, such as cell number, size, each cell distribution, CYP expression, should be clarified.
Response:
We have added the details of the organoids prior to transplantation in the result. About the organoid size and cell distribution, we did measure the organoid size and cell distribution at day 6 and the data were included in result 3.1 and supplement section (page 7, line 318 to 319, figure 1a and page 7, line 320 to 321, supplement Figure S1 a). Because of the using of Ibidi Culture-Insert 2 Well system, the shape of the sheet like-organoid such as length and width has been fixed after 1 day. The mini sheet like-organoids have been merged into a stable rectangular tissue block at day 6 (length: 6.5mm, width: 3.25 mm, height: 100 μm). Unfortunately, we do not have the necessary tool to demonstrate cell numbers at day 6. We tried to digest the rectangular tissue block into single cells to count the cell number at day 6. However, we found that there are some dead cells after digestion, and it is impossible to accurately measure the cell number at day 6.We only show the cell number at day 0 (Materials and Methods 2.3, page 4, line 166 to 170). Our previous experimental data, Liver Bud, which used the Elplasia six-well plates and omni-well-array plates showed the cell number, size and cell distribution of the organoid well [1–4]. Although these methods are suitable for observing the details of organoid in vitro, it is not suitable for implantation over the liver surface in vivo. Our improved method is suitable for in observation in vivo, however, it is a little difficult to obtain comprehensive data in vitro in terms of current technical means. We will try to improve our methods to be applicable to all research in vitro and in vivo. We also found that many researchers reported that the expression levels of CYP450 enzyme in Huh7 during drug metabolizing [5–7]. We will consider testing the expression of CYP in our future drug screening experiments or in the process of anti-tumor treatment.
Reference:
- Camp, J.G.; Sekine, K.; Gerber, T.; Loeffler-Wirth, H.; Binder, H.; Gac, M.; Kanton, S.; Kageyama, J.; Damm, G.; Seehofer, D.; et al. Multilineage communication regulates human liver bud development from pluripotency. Nature 2017, 546, 533–538, doi:10.1038/nature22796.
- Takebe, T.; Sekine, K.; Enomura, M.; Koike, H.; Kimura, M.; Ogaeri, T.; Zhang, R.R.; Ueno, Y.; Zheng, Y.W.; Koike, N.; et al. Vascularized and functional human liver from an iPSC-derived organ bud transplant. Nature 2013, 499, 481–484, doi:10.1038/nature12271.
- Takebe, T.; Zhang, R.-R.; Koike, H.; Kimura, M.; Yoshizawa, E.; Enomura, M.; Koike, N.; Sekine, K.; Taniguchi, H. Generation of a vascularized and functional human liver from an iPSC-derived organbud transplant. Nat. Protoc. 2014, 9, 396–409, doi:10.1038/nprot.2014.020.
- Takebe, T.; Sekine, K.; Kimura, M.; Yoshizawa, E.; Ayano, S.; Koido, M.; Funayama, S.; Nakanishi, N.; Hisai, T.; Kobayashi, T.; et al. Massive and Reproducible Production of Liver Buds Entirely from Human Pluripotent Stem Cells. Cell Rep. 2017, 21, 2661–2670, doi:10.1016/j.celrep.2017.11.005.
- Bulutoglu, B.; Mert, S.; Rey-Bedón, C.; Deng, S.L.; Yarmush, M.L.; Usta, O.B. Rapid maturation of the hepatic cell line Huh7 via CDK inhibition for PXR dependentCYP450 metabolism and induction. Sci. Rep. 2019, 9, 15848, doi:10.1038/s41598-019-52174-w.
- Bulutoglu, B.; Rey-Bedón, C.; Mert, S.; Tian, L.; Jang, Y.-Y.; Yarmush, M.L.; Usta, O.B. A comparison of hepato-cellular in vitro platforms to study CYP3A4 induction. PLoS One 2020, 15, e0229106, doi:10.1371/journal.pone.0229106.
- Choi, S.; Sainz, B.J.; Corcoran, P.; Uprichard, S.; Jeong, H. Characterization of increased drug metabolism activity in dimethyl sulfoxide(DMSO)-treated Huh7 hepatoma cells. Xenobiotica. 2009, 39, 205–217, doi:10.1080/00498250802613620.
- The secretion level of cytokine or chemokine should be investigated.
Response:
In fact, we did perform a preliminary liver tissue cytokine assay and compare the cytokine profiles of the control group, TAA-treated group and the CDAHFD group. Our results showed that there are differences in cytokine profiles between these groups, such as Apo-A1, CRP, IL-1 beta, IL-2, IL-17A, Lipocalin-2, myeloperoxidase, and uPAR. This is the reason in the discussion part we suggested that might be due to the changes of cytokines profiles after HCC implantation. At current stage, our cytokine assay is not conclusive yet. Therefore, we didn’t include the cytokine assay results in the manuscript. We have planned to perform more cytokine tests in our following studies to figure out the mechanism/function of HCC tumorigenesis on cytokines profiles.
- Discussion: Organoid transplantation has already been reported. Therefore, the authors should claim the study comparing the related studies.
Response:
We have added the additional description of organoid transplantation in the discussion: (please see page 20, line 678 to page 21, line 702).
Thank you for your thoughtful feedback. Your valued time is greatly appreciated, and your comments have greatly improved the manuscript.
Round 2
Reviewer 3 Report
All of the comments are responded. I recommend this manuscript to deserve publication in this style.